# Vocabulary-Defined Semantics: Latent Space Clustering for Improving In-Context Learning

## Abstract

In-context learning enables language models (LM) to adapt to downstream data or tasks by incorporating few samples as demonstrations within the prompts. It offers strong performance without the expense of fine-tuning. However, due to the context-length restriction, the demonstrations occupy only a small proportion of usable samples. The limitation exacerbates the difficulty of optimization, since the performance of in-context learning can be unstable depending on the quality, format, or order of demonstrations. Prior work, such as Knn Prompting, index samples based on the similarities of logits at the output-side, in addition to the regular retrieval operation at the input-side. They improve in-context learning by leveraging the core ability of next-token prediction, rather than relying solely on the emergent capacity to make analogies. Despite this, the hard-to-optimize issue of in-context learning still exists. In our view, it stems from the process of selecting demonstrations. To address this, we propose complementing in-context learning with an additional clustering operation, making full use of all usable samples. We propose a novel approach "vocabulary-defined semantics". Grounded in LM vocabulary, which is the label space of model outputs, the proposed approach computes semantically equivalent latent representations for output labels. Then, taking the representations as centroids, a clustering operation is performed to align the semantic properties between the language model and the downstream data/tasks. Based on extensive experiments across diverse textual understanding datasets and multiple models, our approach outperforms the state-of-the-art in terms of effectiveness and efficiency. On average, it achieves $3\% - 49\%$ improvements via the clustering module, while requiring only half of the computation time via the similarity-based logits computation.

## 1 Introduction

Language models (LMs) are drawing significant attention due both to their potential opportunities and associated risks (Bommasani et al., 2021). Despite demonstrating impressive performance, in some scenarios, it may not be convenient to finetune them to downstream data/tasks, such as Language Model-as-a-Service (LMaaS) (Sun et al., 2022). As a solution, in-context learning (Dong et al., 2022) serves as an effective approach to utilizing language models for downstream tasks. Unlike model fine-tuning, in-context learning adapts to downstream data or tasks by incorporating demonstrations into the prompts. Using the emerging ability to draw analogies from demonstrations (Wei et al., 2022), it offers strong performance without the expense associated with model fine-tuning.

However, in-context learning is not a stable technique and may be affected by multiple factors, which exacerbates the difficulty of further optimization. For example, the performance may be affected by the quality of demonstrations (Rubin et al., 2021; Li et al., 2023), their format (Yang et al., 2023a;b), and even their order (Liu et al., 2024; Guo et al., 2024). The state-of-the-art (SOTA) KNN prompting (Xu et al., 2023) takes a significant step forward. Rather than solely depending on the emergent ability of language models to make analogies, the solution also utilizes the core capability of next-token prediction to enhance in-context learning. In addition to the retrieval operation of selecting demonstrations at the input-side, it introduces an additional indexing operation to support KNN decision at the output-side. The limitations are, while using the indexed data as the reference in KNN decision, it cannot cooperate with the normal LM inference. Also, it cannot handle the case where the samples corresponding to a specific output label are unavailable.

Nevertheless, the critical issue of in-context learning still exists. In our view, it is caused by the drawbacks of selecting demonstrations (also applies to the case of composing demonstrations). First, in-context learning retrieves only a subset from all usable demonstrations, so the retrieval operation tends to be a suboptimal option, since it is unclear which samples would be the best choices. Second, assume the best samples to retrieve can be known beforehand, the retrieval operation is still suboptimal, since the best samples may not included in the retrievable samples. To solve the issue of difficult optimization, we complement in-context learning with an additional clustering operation, making full use of all usable samples. Similar to SOTA KNN Prompting (Xu et al., 2023), our focus is on the output-side, not the input-side. Meanwhile, we replace its indexing operation with a clustering operation that aligns semantic properties, to overcome the limitations.

We propose a novel and effective approach: *Vocabulary-Defined Semantics*, short for *VDS*. Grounded in LM vocabulary, which is the label space of model outputs, we compute the semantically equivalent latent representations for all output labels. Then, we determine the logits directly in latent space (which is disentangled for latent representations), and use the logits to quantify the semantic gap between the language model and downstream data, so we will know on the fly how good the latent representations are. Last, taking the representations as centroids, we cluster the representations of downstream data to mitigate their semantic gap with the language model. By defining semantic bases of the label space, computing disentangled logits to quantify the semantic gap, and clustering representations to optimize the logits, our approach incorporates KNN decisions into LM inference. It aggregates the semantic information from the downstream data and cooperates with LM inference.

In addition, we have conducted large-scale experiments and detailed analysis across diverse datasets of text understanding, with four model families: GPT2 (0.13B-1.61B) (Radford et al., 2019), Qwen2 (0.49B-7.62B) (Yang et al., 2024), Gemma2 (2.61B-9.24B) (Riviere et al., 2024), and Llama3 (8.03B) (Dubey et al., 2024). Based on our results, our approach outperforms the SOTA in both effectiveness and efficiency. On average, our approach obtains $3\% - 49\%$ improvements (via the clustering module), while taking only half of the computation time (via the similarity-based logits computation). The replication repository is attached as supplementary material.

Our contributions are as follows:

- We define a collection of specialized representations, termed "semantic bases" for disentangled semantics. It allows for more precise alignment of semantic properties during tasks like language model inference, leading to improved performance and stability.

- We propose a novel way to compute logits via similarity measurement instead of common matrix multiplication, leveraging the local isotropy of LM latent space. The logits can be used to quantify the semantic gap between the language model and downstream data.

- We implement semantic-based clustering on latent representations through a lightweight neural module. The clustering operation aligns the semantic property of LM latent space with downstream data. It makes full use of all usable data to improve the performance of in-context learning, while reducing the computation cost by orders of magnitude.

## 2 BACKGROUND

### 2.1 IN-CONTEXT LEARNING

In-context learning (ICL) is a prompting paradigm of language models. It retrieves samples and includes them as demonstrations in the prompts, to let models learn from the analogy (Dong et al., 2022). In the basic usage of language models, a model shall predict an output $y$ for the given input $x$. In contrast, ICL will retrieve samples from a corpus of all usable input-output pairs, to obtain a collection $\mathcal{T} = \{(x_i, y_i)\}$. The retrieved samples will be filled in templates $\pi(x_i, y_i)$ and concatenate with the given input $x$ as a prompt: $\pi(x_1, y_1) \oplus \pi(x_2, y_2) \oplus \ldots \oplus \pi(x_{|\mathcal{T}|}, y_{|\mathcal{T}|}) \oplus \pi(x, *)$. Then, the model will turn to predict an output $y$ for the constructed prompt. Benefit from the emergent abilities of large-scale LMs, the use of in-context learning can avoid the expensive cost of model finetuning, while obtaining the equivalent performance in the adaptation to downstream data or tasks.

## 2.2 ENTANGLEMENT AND DISENTANGLEMENT

Entanglement indicates the phenomenon where the elements of a vector data correlate with each other. In contrast, disentanglement means that the elements are independent of each other (Higgins et al., 2018). For example, LM latent representations, namely distributed representations (such as $[0.1, 0.3, 0.4, 0.2]$), are entangled while onehot embeddings (such as $[0, 0, 1, 0]$) are disentangled.

Following the definitions, for the representations in latent space, the logits computed on the vocabulary is entangled. Recall the computation in language models, in the forward-pass, the logits is computed on LM vocabulary and compared with the ground truth to obtain the loss. Then, in the backward-pass, the loss is backpropagated to model layers to obtain the gradients for model parameters. The dimension size of latent space is smaller than the vocabulary size, and the change in dimension size indicates the meaning of each element is no longer independent, which indicates an entanglement. Therefore, the logits is entangled for LM latent space even though disentangled for LM vocabulary. The entanglement is caused by the dimension transformation, namely the LM-head matrix.

Entanglement tends to restrict the potential improvements in efficiency and robustness. An entangled logits cannot represent a clear and intuitive meaning in latent space, which will cause challenges to further optimization. On efficiency, the dimension transformation via a huge LM-head matrix is mandatory in logits computation, so it is hard to reduce the costly computation; On robustness, the entangled logits will be sensitive to the absolute numerical magnitude of latent representations, which damages flexibility. Or else, the precision may not be guaranteed. In this paper, leveraging the disentangled logits, our approach obtains breakthroughes on these restrictions.

## 3 VOCABULARY-DEFINED SEMANTICS

Our approach defined and utilized the semantics of LM latent space, leveraging the vocabulary, as illustrated in Figure 1. We name our approach Vocabulary-Defined Semantics, short as VDS. In VDS: (1) First, we obtain the semantically equivalent latent representations of vocabulary labels by solving the pseudoinverse of the LM-head matrix, termed as "semantic basis". Semantic bases faithfully represent the label space of model outputs and can cover all output labels in the vocabulary; (2) Then, based on the local isotropy of LM latent space (Cai et al., 2021), we can compute logits by measuring the similarities between latent representations and semantic bases. The novel practice of logits computation is directly in the latent space. Therefore, the computed logits is disentangled to latent representations, and can be used to quantify the semantic gap between the language model and downstream data; (3) Further, taking semantic bases as centroids, we perform centroid-known clustering on the latent representations of usable samples. The clustering operation can optimize the logits and mitigate the semantic gap of the language model with the downstream data.

### 3.1 VOCABULARY-DEFINED SEMANTIC FIELDS

In the common way of logits computation, the data representation is projected to the LM vocabulary, and its actual semantic meaning is represented by the probability distribution on the vocabulary. Given this, we use the vocabulary to define the semantic property of LM latent space. Considering onehot embeddings are the most semantically representative distributions in the vocabulary, we compute the corresponding latent representations as the semantically equivalents of vocabulary labels. Since the softmax operation maintains monotonicity, onehot embeddings shall be regarded as logits.

For a given LM-head matrix, we conduct matrix multiplication to get the corresponding representation in the latent space. Since the computation direction is from logits to representations, instead of using the LM-head matrix $\mathbb{W}$, we use its pseudoinverse $\mathbb{W}^+$. If there are $v$ labels in the vocabulary, there will be $v$ unique semantic bases representing all semantic meanings. At the output side of LM, we multiply each onehot embedding $\vec{e}$ by the pseudoinverse matrix $\mathbb{W}^+$ to obtain the corresponding representation $\vec{s}$. That is, $\vec{s} = \vec{e} \cdot \mathbb{W}^+$. The computation is equivalent to solving the least squares problem of a system of linear equations. We call each of the obtained vectors semantic basis.

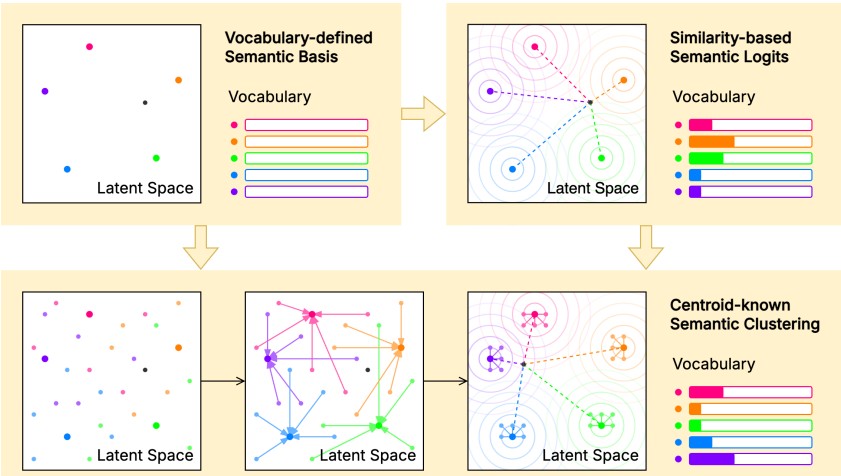

Figure 1: We illustrate with an LM, where the vocabulary consists of five colorful labels: (1) On the semantic basis (upper-left), we obtained the semantic bases in the latent space, represented by large color dots. The colors correspond to the labels in the vocabulary one to one; (2) On the semantic logits (upper-right), the small dark dot means the representation of arbitrary data. We measure its similarities with the semantic bases as logits. The logits are normalized as probabilities on the vocabulary, and the argmax label is orange; (3) On the semantic clustering (bottom), small color dots mean the representations of usable samples, and their colors indicate the corresponding ground truth. Originally, they were mixed and scattered all around. By clustering, they are gathered by color and into clusters centered in semantic bases. The clustering operates on the latent space, affecting the dark dot as well. The logits of the dark has changed, and its argmax label becomes violet.

## 3.2 SIMILARITY-BASED SEMANTIC LOGITS

We directly determine the logits in the LM latent space, not on the vocabulary. In common practice, the logits are computed after the LM-head. In the LM forward-pass, the logits is obtained by the multiplication between the last-layer latent representation and the LM-head matrix, and then is compared with the ground truth to compute the loss. Further, in the backward-pass, the loss will propagate the gradients back from the LM-head to the embeddings layer. In our approach, the logits is obtained before the LM-head. We use the similarity between latent representation and semantic bases to compute the logits. The similarity-based logits is disentangled to latent representations. It becomes intuitive than the disentangled logits, and will support further operations.

The latent space of Transformer models is proved to be locally isotropic (Cai et al., 2021), in terms of information and semantics. Therefore, we can compute logits in latent space by measuring the distances between representations, as an alternative to the typical way of matrix multiplication on the vocabulary. Isotropy means that the properties of a space are uniform in all directions. That is, in different directions of the latent space, the semantic differences of representations in the same distance remain close. Therefore, when we consider all semantic bases, instead of only one basis, the hypothesis is almost true that *the similarity of representations in the latent space remains positively correlated with the similarity of the corresponding distributions on the vocabulary*.

---

**Algorithm 1:** Similarity-based Semantic Logits

---

**Data:** $n$ semantic bases $\vec{b_i}$; representation $\vec{r}$
**Result:** probability distribution $probs$

1 $logits \leftarrow \texttt{init\_1d\_tensor}(\underline{n})$;
2 **for** $i \leftarrow 0$ **to** $n-1$ **do**
3     $logits\,[i] \leftarrow \texttt{similarity\_metric}(\vec{b_i}, \vec{r})$;
4 $probs \leftarrow \texttt{softmax}(\underline{logits})$;

---

*Similarity-based Semantic Logits.* For a given representation, the logits is commonly computed by multiplying with the LM-head matrix, a novel practice is to compute its projection to the semantic bases using cosine similarity. Leveraging the positive correlation between the representation similarity in the latent space and the distribution similarity on the vocabulary, we compute the logits based on their similarities with the semantic bases. The similarity metric tends to be the cosine similarity, so the logits indicate the projections to the semantic bases. Further, the logits can be used to compute the probability distribution on the vocabulary, as shown in Algorithm 1.

Based on similarity-based logits, the difference between the normal LM inference and KNN decision can be reformulated as the semantic gap of semantic bases (namely the language model) and downstream data. The effect of similarity-based logits is equivalent to that in normal LM inference. Taking the downstream data to replace semantic bases in logits computation, the effect of logits will make LM inference similar to KNN decision. The difference in results of normal LM inference and KNN decision indicates a semantic gap. Further, by mitigating the semantic gap, we will incorporate KNN decisions into LM inference to guarantee the adaptation of in-context learning.

### 3.3 CENTROID-KNOWN SEMANTIC CLUSTERING

The semantic gap of KNN decision to the normal LM inference can be explained as, the semantic logits of downstream data cannot correspond to the ground truth. As a solution, we cluster the representations of downstream data in the last-layer latent space, to let them be closer to the ground truth, namely the corresponding semantic bases. Through centroid-known clustering on downstream data, the logits can quantify the semantic gap between the language model and the downstream data. The objective of clustering is to optimize the logits, that is, mitigating the semantic gap.

We specify each semantic basis as the centroid of each cluster. In the optimal situation, the last-layer representations of the same semantic meaning (corresponding to the same label), should stay in the same cluster. It can be taken by non-neural methods. We adopt a learning-based neural clustering method, so we can study more on similarity-based logits (in Section 5).

$$\lambda\left(\vec{r}\right) = \text{LN}\left(\text{MLP}\left(\vec{r} \odot \text{CA}\left(\vec{r}\right)\right)\right) \tag{1}$$

$$\text{CA}\left(\vec{r}\right) = \text{Bn}\left(\text{avg}\left(\vec{r}\right)\right) + \text{Bn}\left(\text{max}\left(\vec{r}\right)\right) \tag{2}$$

We introduce a simple neural clustering module for semantic clustering, which consists of a channel attention (`CA`), a multi-layer perceptron (`MLP`), and a layer normalization (`LN`), as shown in Equation (1). The channel attention learns the coefficients of the representations, based on the information in the channel domain (Hu et al., 2017), as shown in Equation (2). The coefficients are a sum of the Bottleneck (`Bn`) outputs, respectively on the average-pooling (`avg`) and the maximum-pooling (`max`) of representations. Meanwhile, `MLP` and `LN` learn non-linear transformations in the latent space.

*Logits-Numerical Sensitivity.* In latent space clustering, whether the logits is entangled will affect the performance. A disentangled logits shall indicate a more robust logits-numerical sensitivity. We use the term to describe the sensitivity of loss computation to the numeric values of logits.

## 4 EXPERIMENTS

In the following, we will introduce the experimental setup, and discuss the results on the effectiveness and efficiency. Following the prior work of in-context learning, the experiments are conducted with a wide range of model scales on diverse datasets, as shown in Table 1 and Table 2. On the results, the optimal scores are stressed with gray color.

### 4.1 SETUP

*Baselines.* In our experiments, we study the effects of semantic clustering on general in-context learning, short as ICL (Min et al., 2022). Besides, we compare with the SOTA in-context learning method, $K$NN prompting, short as KP (Xu et al., 2023). We reproduce the practice of ICL and KP following the paper (Xu et al., 2023). As an extension of the study, we also compare with parameter-efficient finetuning methods, including LoRA (Hu et al., 2021) and IA³ (Liu et al., 2022).

| | GPT2 | | | | Qwen2 | | | Gemma2 | | Llama3 |
|---|---|---|---|---|---|---|---|---|---|---|
| | **Small** | **Medium** | **Large** | **XL** | **0.5B** | **1.5B** | **7B** | **2B** | **9B** | **8B** |
| Parameters | 137M | 380M | 812M | 1.61B | 494M | 410M | 7.62B | 2.61B | 9.24B | 8.03B |
| Num. of Layer | 12 | 24 | 36 | 48 | 24 | 28 | 28 | 26 | 42 | 32 |
| Dimension Size | 768 | 1,024 | 1,280 | 1,600 | 896 | 1,536 | 3,584 | 2,304 | 3,584 | 4,096 |
| Vocabulary Size | 50,257 | | | | 151,936 | | 152,064 | 256,000 | | 128,256 |

Table 1: Stats of Language Models. We use recognized open-source LMs and involve four model families in our experiments: GPT2 (0.13B-1.61B) (Radford et al., 2019), Qwen2 (0.49B-7.62B) (Yang et al., 2024), Gemma2 (2.61B-9.24B) (Riviere et al., 2024), and Llama3 (8.03B) (Dubey et al., 2024).

| | | AGNews | CARER | MR | MRPC | SST5 | SUBJ | TREC | WebSS |
|---|---|---|---|---|---|---|---|---|---|
| Num. of Classes | | 4 | 6 | 2 | 2 | 5 | 2 | 6 | 8 |
| Class Balance | | ✓ | ✗ | ✓ | ✗ | ✗ | ✓ | ✗ | ✗ |
| Avg. Length of Prompts | | 59.2 | 26.1 | 31.9 | 63.2 | 29.0 | 34.1 | 16.4 | 29.9 |
| Num. of Data | Train | 120,000 | 16,000 | 8,662 | 4,076 | 8,544 | 8,000 | 5,452 | 10,060 |
| | Test | 7,600 | 2,000 | 2,000 | 1,725 | 2,210 | 2,000 | 500 | 2,280 |
| Num. of Shots | 1k-context | 2 | 4 | 8 | 4 | 4 | 8 | 8 | 2 |
| | 2k-context | 4 | 8 | 16 | 8 | 8 | 16 | 16 | 4 |

Table 2: Stats of Datasets, and Number of Allowed Shots in the Prompts. We conduct evaluations with established text understanding tasks, respectively for text classification: AGNews (Zhang et al., 2015), SUBJ (Pang & Lee, 2004), TREC (Voorhees & Tice, 2000), and WebSS (Phan et al., 2008); sentiment analysis: MR (Pang & Lee, 2005) and SST5 (Socher et al., 2013); emotion recognition: CARER (Saravia et al., 2018); as well as similarity detection: MRPC (Dolan & Brockett, 2005).

*Metrics.* We follow the common practice, using accuracy to evaluate balanced datasets, and F1 to evaluate imbalanced datasets. The reason is that, accuracy is preferred for its simplicity and directness, while F1 shall be taken when the precision-recall trade-off is a concern.

*Environments.* Our implementation is based on PYTORCH (Paszke et al., 2019) and TRANSFORM-ERS (Wolf et al., 2019). The experiments are conducted on Nvidia V100 GPU (32 GB). In the experiments with Qwen2, Gemma2, and Llama3 models, we apply INT4 quantization to reduce the memory cost (while maintaining the model performance) (Dettmers et al., 2023; Wu et al., 2023a). The LM quantization is dependent on the PEFT [1] library. GPT2 models are loaded without quantization.

### 4.2 PIPELINES

We use few-shot prompts for in-context learning methods, making full use of the allowed context length. The prompt templates used in our experiments are available in Appendix A. For ICL, we retrieve the demonstrations randomly from the training set, and keep the number of demonstrations the same for each class. For KP, the retrieval operation is the same. The additional indexing operation follows the official implementation, which is based on KL divergence. The downstream data that is more similar in the probability distribution are more likely to be retrieved. For PEFT methods, we use zero-shot prompts to reduce the effects of in-context prompting. Meanwhile, we specify the recommended modules as trainable based on their papers. That is, the QKV matrices in LORA are trainable; QKV matrices plus the second FC layer in $IA^3$) are trainable.

### 4.3 EFFECTIVENESS STUDY

Based on the results shown in Table 3, on average, VDS shows the optimal performance, while KP shows the suboptimal performance, both are better than ICL. Comparing the improvements of VDS on different language models, the improvements on Gemma2 models are most significant, which are 44% and 49%; while the improvements on Llama3 are marginal, which is 3%. The mean value of average improvements is 15.3%. The difference is mainly decided by the parameter amount of

---

[1] https://github.com/huggingface/peft

| Dataset | Method | GPT2 | | | | Qwen2 | | | Gemma2 | | Llama3 | Avg. |
|---|---|---|---|---|---|---|---|---|---|---|---|---|
| | | Small | Medium | Large | XL | 0.5B | 1.5B | 7B | 2B | 9B | 8B | |
| AGNews (Acc.) | ICL | 0.302 | 0.571 | 0.533 | 0.752 | 0.816 | 0.802 | 0.809 | 0.250 | 0.250 | 0.876 | 0.596 |
| | KP | 0.867 | 0.875 | 0.869 | 0.874 | 0.866 | 0.883 | 0.892 | 0.258 | 0.250 | 0.898 | 0.753 |
| | VDS | 0.918 | 0.924 | 0.928 | 0.928 | 0.903 | 0.938 | 0.939 | 0.907 | 0.907 | 0.940 | 0.923 |
| CARER (F1) | ICL | 0.133 | 0.086 | 0.130 | 0.098 | 0.229 | 0.337 | 0.484 | 0.053 | 0.092 | 0.371 | 0.201 |
| | KP | 0.323 | 0.322 | 0.317 | 0.283 | 0.518 | 0.466 | 0.477 | 0.136 | 0.075 | 0.500 | 0.342 |
| | VDS | 0.613 | 0.607 | 0.589 | 0.643 | 0.738 | 0.680 | 0.700 | 0.653 | 0.630 | 0.627 | 0.648 |
| MR (Acc.) | ICL | 0.501 | 0.500 | 0.705 | 0.526 | 0.843 | 0.907 | 0.936 | 0.500 | 0.500 | 0.936 | 0.685 |
| | KP | 0.730 | 0.723 | 0.831 | 0.843 | 0.852 | 0.900 | 0.924 | 0.497 | 0.500 | 0.923 | 0.772 |
| | VDS | 0.810 | 0.855 | 0.859 | 0.866 | 0.852 | 0.903 | 0.929 | 0.737 | 0.726 | 0.906 | 0.844 |
| MRPC (F1) | ICL | 0.399 | 0.399 | 0.399 | 0.399 | 0.399 | 0.399 | 0.399 | 0.399 | 0.399 | 0.399 | 0.399 |
| | KP | 0.543 | 0.515 | 0.586 | 0.558 | 0.608 | 0.676 | 0.695 | 0.489 | 0.251 | 0.609 | 0.553 |
| | VDS | 0.620 | 0.627 | 0.631 | 0.633 | 0.615 | 0.729 | 0.785 | 0.616 | 0.605 | 0.641 | 0.650 |
| SST5 (F1) | ICL | 0.082 | 0.158 | 0.075 | 0.149 | 0.308 | 0.404 | 0.485 | 0.115 | 0.114 | 0.351 | 0.224 |
| | KP | 0.322 | 0.356 | 0.382 | 0.384 | 0.398 | 0.450 | 0.482 | 0.175 | 0.089 | 0.447 | 0.349 |
| | VDS | 0.392 | 0.451 | 0.436 | 0.459 | 0.441 | 0.483 | 0.526 | 0.347 | 0.369 | 0.456 | 0.436 |
| SUBJ (Acc.) | ICL | 0.500 | 0.501 | 0.502 | 0.559 | 0.670 | 0.930 | 0.843 | 0.500 | 0.500 | 0.933 | 0.644 |
| | KP | 0.807 | 0.868 | 0.889 | 0.900 | 0.853 | 0.940 | 0.940 | 0.525 | 0.500 | 0.947 | 0.817 |
| | VDS | 0.914 | 0.923 | 0.931 | 0.945 | 0.927 | 0.952 | 0.959 | 0.894 | 0.905 | 0.961 | 0.931 |
| TREC (F1) | ICL | 0.472 | 0.414 | 0.572 | 0.436 | 0.772 | 0.718 | 0.861 | 0.053 | 0.038 | 0.787 | 0.512 |
| | KP | 0.818 | 0.816 | 0.872 | 0.833 | 0.850 | 0.901 | 0.914 | 0.089 | 0.157 | 0.873 | 0.712 |
| | VDS | 0.896 | 0.924 | 0.916 | 0.914 | 0.907 | 0.925 | 0.943 | 0.890 | 0.894 | 0.892 | 0.910 |
| WebSS (F1) | ICL | 0.245 | 0.132 | 0.096 | 0.209 | 0.349 | 0.397 | 0.415 | 0.034 | 0.029 | 0.436 | 0.234 |
| | KP | 0.704 | 0.712 | 0.628 | 0.705 | 0.775 | 0.832 | 0.805 | 0.102 | 0.029 | 0.835 | 0.613 |
| | VDS | 0.796 | 0.806 | 0.793 | 0.803 | 0.785 | 0.815 | 0.836 | 0.731 | 0.727 | 0.828 | 0.792 |
| **Avg.** | ICL | 0.329 | 0.345 | 0.377 | 0.391 | 0.548 | 0.612 | 0.654 | 0.238 | 0.240 | 0.636 | 0.437 |
| | KP | 0.639 | 0.648 | 0.672 | 0.673 | 0.715 | 0.756 | 0.766 | 0.284 | 0.231 | 0.754 | 0.614 |
| | VDS | 0.745 | 0.765 | 0.760 | 0.774 | 0.771 | 0.803 | 0.827 | 0.722 | 0.720 | 0.781 | 0.767 |

Table 3: Performance on Text Understanding with In-Context Learning Methods.

language models. By comparing the models of similar size (namely Qwen2-7B, Gemma2-9B, and Llama3-8B), where Gemma2-9B has a doubled vocabulary size to Llama3-8B, we may conclude that a large vocabulary size indicates better improvements. It fulfills the intuition, since a large vocabulary indicates the semantic meanings can be better distinguished via clustering. In other words, when the vocabulary is larger, the existing methods are more likely to underperform while our "vocabulary-defined semantics" tends to show better advantages. In contrast, the dimension size is not a critical factor since the difference is marginal, so the high-dimensional property of latent space shall be very similar. In addition, the quality of latent representations in different models also matters.

Meanwhile, as shown in Table 3, compared with ICL and KP, our approach shows huge improvements in CARER, TREC, and WebSS datasets, which are 45%, 40% and 56% compared to ICL meanwhile 31%, 20% and 18% compared to KP; then show moderate improvements in AGNews and SUBJ datasets; and last show slight advantages in MR, MRPC, and SST5 datasets. The degree of improvements varies on datasets while the causes are not the obvious factors in the experiments, such as the number of shots in the prompts, the amount of usable downstream data, etc. Instead, the improvements are mainly affected by the task. In text classification and emotion recognition (CARER), our "vocabulary-defined semantics" can show great advantages over others; but in sentiment analysis (MR, SST5) and similarity detection (MRPC), the improvements seem not that obvious. Based on the details of datasets, the output labels in MR are easy to distinguish while in SST5 are challenging. Therefore, for MR, ICL and KP are good enough so the space for improvements is small; while for SST5, the quality of latent representations may not be that good, therefore, the scores on SST5 are the lowest. In addition, MRPC requires the reasoning ability of language models, which explains why the improvements are not that good. Our semantic-based approach is dependent on the quality of latent representations, this is the reason why VDS performs better on other datasets.

Based on the results shown in Table 4, in-context learning methods, including our approach, perform better than LoRA but worse than IA[3]. It indeed indicates a performance gap between in-context learning methods and finetuning methods, while the gap is accompanied by the additional computation cost, as well as the suitability in different usage scenarios. Besides, the reason why LoRA shows bad

| Method | AGNews (Acc.) | CARER (F1) | MR (Acc.) | MRPC (F1) | SST5 (F1) | SUBJ (Acc.) | TREC (F1) | WebSS (F1) | Avg. |
|--------|-------|--------|------|-------|------|------|------|-------|------|
| LoRA | 0.272 | 0.075 | 0.500 | 0.399 | 0.075 | 0.842 | 0.430 | 0.174 | 0.346 |
| IA$^3$ | 0.941 | 0.889 | 0.944 | 0.727 | 0.464 | 0.979 | 0.940 | 0.916 | 0.850 |
| ICL | 0.876 | 0.371 | 0.936 | 0.399 | 0.351 | 0.933 | 0.787 | 0.436 | 0.636 |
| KP | 0.898 | 0.500 | 0.923 | 0.609 | 0.447 | 0.947 | 0.873 | 0.835 | 0.754 |
| **VDS** | 0.940 | 0.627 | 0.906 | 0.641 | 0.456 | 0.961 | 0.892 | 0.828 | 0.781 |

Table 4: Performance on Text Understanding with Llama3-8B with PEFT methods.

performance is because LoRA only finetune the self-attention module but the feed-forward module tends to be the better choice in model adaptation (Geva et al., 2020; 2022; Hassid et al., 2022).

## 4.4 EFFICIENCY STUDY

On the computation cost, KP requires the most, then PEFT methods, while ICL and VDS require the least. As shown in Table 5, based on the average time cost, ICL is the fastest and only costs around 1.3 hours. Then, VDS and KP take 2.3 and 5.4 hours respectively, while LoRA and IA$^3$ cost around 3.6 hours. The INFERENCE operation of in-context learning methods takes longer time, since ICL and KP are using few-shot prompts, and the prompts occupy the 1k-context (GPT2 LMs) and 2k-context (Pythia LMs). In contrast, PEFT methods and VDS take zero-shot prompts so their inference is faster. Except from ICL, besides INFERENCE, all methods have an additional operation. The indexing operation of KP finds high-quality demonstrations for few-shot prompting by analyzing the logits and the ground truth of training data. It consists of a forward-pass with few-shot prompts and the ranking of neighboring logits (measured by KL divergence). PEFT methods require LM forward-pass and backward-pass with zero-shot prompts, the amount of trainable parameters is small but the computation cost of backpropagation cannot be reduced. In contrast, VDS requires LM forward-pass but the backward-pass with zero-shot prompts is only on the neural clustering module. The clustering operation of VDS is on usable downstream data, since the data amount of AGNews is larger than other datasets, the time cost of clustering on AGNews is greatly larger than on other datasets. In addition, due to the disentanglement, the time cost of logits computation can be further reduced to $1/v$ by merely optimizing with the ground truth, without damaging the performance.

| Method | Operation | AGNews | CARER | MR | MRPC | SST5 | SUBJ | TREC | WebSS | Avg. |
|--------|-----------|--------|-------|-----|------|------|------|------|-------|------|
| LoRA | Training | 19.776 | 2.112 | 1.205 | 0.693 | 1.152 | 1.123 | 0.679 | 1.336 | 3.510 |
| | Inference | 0.440 | 0.093 | 0.098 | 0.104 | 0.108 | 0.100 | 0.023 | 0.107 | 0.134 |
| IA$^3$ | Training | 19.605 | 2.095 | 1.190 | 0.685 | 1.144 | 1.118 | 0.673 | 1.328 | 3.480 |
| | Inference | 0.437 | 0.092 | 0.098 | 0.104 | 0.105 | 0.099 | 0.023 | 0.107 | 0.133 |
| ICL | Inference | 3.471 | 1.408 | 1.020 | 0.908 | 1.278 | 1.051 | 0.395 | 1.205 | 1.342 |
| KP | Indexing | 2.067 | 3.279 | 1.114 | 1.075 | 2.954 | 1.063 | 3.800 | 3.773 | 2.391 |
| | Inference | 8.841 | 3.005 | 1.744 | 1.486 | 3.027 | 1.709 | 0.776 | 3.864 | 3.056 |
| **VDS** | Clustering | 11.370 | 1.405 | 0.766 | 0.405 | 0.754 | 0.711 | 0.477 | 0.881 | 2.096 |
| | Inference | 0.261 | 0.176 | 0.176 | 0.168 | 0.193 | 0.176 | 0.043 | 0.199 | 0.174 |

Table 5: Runtime on Text Understanding with Llama3-8B (in Hours).

On the storage cost, KP requires the most storage cost while PEFT methods and VDS have much lower costs. For a given language model, assume its dimension size of latent representations is $d$, the number of LM layers is $l$, and the size of LM vocabulary is $v$. ICL has no additional storage. KP needs to store the logits of neighboring prompts from the usable data, for each label in the vocabulary, that is, $k * v * v$ when the number of nearest neighbors is $k$. As PEFT methods, LoRA and IA$^3$ store the additional trainable parameters. Taking their recommended setups, the amount of their parameters are $4 * r * d * l$ and $7 * d * l$, when the low-rank parameter is $r$. For our approach VDS, the storage cost is the neural clustering module, that is, $\frac{33}{8} * d * d$. The number of trainable parameters consists of $4 * d * d$ for MLP and $\frac{1}{8} * d * d$ for CA. It can be further reduced to $\frac{33}{8} * d$ via parameterized hypercomplex multiplication (Zhang et al., 2021).

| Category | Variant | AGNews (Acc.) | CARER (F1) | MR (Acc.) | MRPC (F1) | SST5 (F1) | SUBJ (Acc.) | TREC (F1) | WebSS (F1) | Avg. |
|---|---|---|---|---|---|---|---|---|---|---|
| | **VDS[sm][sm]** | 0.940 | 0.627 | 0.906 | 0.641 | 0.456 | 0.961 | 0.892 | 0.828 | 0.781 |
| Inference | VDS[sm][mm] | 0.940 | 0.627 | 0.906 | 0.641 | 0.459 | 0.960 | 0.892 | 0.830 | 0.782 |
| Clustering | VDS[mm][sm] | 0.934 | 0.645 | 0.906 | 0.647 | 0.449 | 0.964 | 0.914 | 0.804 | 0.783 |
| | VDS[sm-gt][sm] | 0.939 | 0.653 | 0.912 | 0.661 | 0.456 | 0.965 | 0.916 | 0.819 | 0.790 |
| Disentangle | VDS[mm-exp][sm] | 0.250 | 0.086 | 0.500 | 0.399 | 0.075 | 0.500 | 0.038 | 0.029 | 0.235 |
| | VDS[sm-exp][sm] | 0.940 | 0.640 | 0.909 | 0.638 | 0.437 | 0.953 | 0.906 | 0.805 | 0.779 |

Table 6: Performance of Ablation Study with Llama3-8B Model.

## 5 ANALYSIS AND EXPLANATION

In the following, we take an ablation study to further discuss similarity-based logits computation. It is centered on the effects of logits on the performance, when taking different practices of logits computation in clustering or inference operations. We will discuss more on logits disentanglement as well. The experiments are conducted on the same textual datasets, using Llama3-8B model. We use labels in the form of VDS[CLUSTER][INFER] to distinguish different settings, where CLUSTER and INFER indicate the practice of logits computation in clustering or inference operations.

*Logits Computation.* We include VDS as the baseline, marked as VDS[sm][sm], where the similarity-based logits computation (short as sm) is used in both operations of inference and clustering. By comparing with the common way of using matrix multiplication (short as mm) for logits computation in inference, we can see how similarity-based logits make effects in forward-pass, so prepare such a variant VDS[sm][mm]; By comparing with the common logits computation in clustering, we can see how similarity-based logits differ in backward-pass, prepare such a variant VDS[mm][sm]. Meanwhile, we prepare a variant VDS[sm-gt][sm] to validate that the cost in computing similarity-based logits can be further optimized to $1/v$. It merely measures the similarity of the latent representation with one semantic basis, that is the corresponding ground truth (short as gt), instead of all semantic bases.

*Logits Disentanglement.* By amplifying and suppressing the logits values before loss computation, we can show the difference between entangled logits and disentangled logits, and further, reveal the effects of logits disentanglement. We take a simple strategy to manipulate the logits, that is, applying the natural exponential function (short as exp) to the logits, and then, normalizing the results. This operation amplifies the largest values and suppresses small values, while maintaining the magnitude relation of values. Therefore, we prepare such variants VDS[mm-exp][sm] and VDS[sm-exp][sm], for matrix multiplication and similarity measurement respectively.

Based on the results shown in Table 6, compared to the baseline approach VDS, both VDS[sm][mm] and VDS[mm][sm] show similar performance on average. The minor difference between the performance of VDS and two variantsmm indicates an almost equivalence of two practices of logits computation in LM forward-pass, in terms of the effects, no matter whether for either inference or clustering. It proves the correctness of our hypothesis on the correlations of the representations similarity and distribution similarity. Meanwhile, we found in some datasets (AGNews, SST5, WebSS), VDS[mm][sm] performs not as good as the baseline, while in other datasets performs better. It indicates the two practices are not strictly equivalent, and their differences are more likely to cause slight differences in clustering than in inference. Further, we can know that cosine similarity may not be the optimal metric for similarity-based logits, if the two practices must be strictly equivalent.

In addition, the performance of VDS[sm-gt][mm] is slightly better than that of VDS. This is because the variant is better utilizing the logits disentanglement to avoid the effects of these irrelevant semantic bases in clustering. Meanwhile, their computation costs differ a lot. Taking Llama3-8B as an example, the vocabulary size is $128256$ and the dimension size is $4096$, so each time of logits computation, we need to measure the similarity with all semantic bases. The required floating-point operations is $3.15 \times 10^9$ FLOPs. It is more costly than using matrix multiplication for logits computation, which only requires $1.05 \times 10^9$ FLOPs. In contrast, VDS[sm-gt][mm] only measures the similarity with one semantic basis in logits computation, which only requires $2.46 \times 10^4$ FLOPs. It means a **five-order-of-magnitude** improvement in the computational cost in each logits computation, which is a huge computation advantage of similarity-based logits.

On logits disentanglement, compared with VDS[mm][sm], the performance of VDS[mm–exp][sm] shows a huge performance drop. In contrast, compared with VDS, the performance of VDS[sm–exp][sm] is almost the same. The reason explaining the contrasting phenomena is the logits disentanglement. For matrix-multiplication logits, the logits is entangled so it is sensitive to the value differences on vocabulary labels. In contrast, for similarity-based logits, the logits is disentangled so it shows better robustness, and is insensitive to the value differences on vocabulary labels. A robust logits-numerical sensitivity supports direct and complex manipulations on the latent representations or even on the logits, since the loss computation is only sensitive to the relative magnitude for numeric values of logits, instead of the absolute magnitude.

## 6 RELATED WORK

Generally, the performance of in-context learning depends on the data retrieved to prompt language models. The retrieved data is often the task demonstrations, which are in the form of an input-output pair, and will compose with the given input as the new input. In prior work, such as self-adaptive in-context learning (Wu et al., 2023b), the selection and ordering of data retrieval are optimized to improve the model performance. The retrieval is based on the similarity between the embeddings of the corpus data and the representations of the given input. However, KP also retrieve data at the output side, such as the LM last-layer, computing the similarities based on the logits. The data to retrieve is key-value pairs, where the representation is the key and the corresponding ground truth is the value. Then LM does inference with hybrid logits: one is from the model prediction while the other one is from the KNN decision. The hybrid logits will be used to compute a normalized probability distribution. It has been used for language modeling in $k$NN-LM (Khandelwal et al., 2020) and machine translation in $k$NN-MT (Khandelwal et al., 2021). A similar practice is using activations as the key (Grave et al., 2017). A further topic along this direction is to realize retrieval-based neuro-symbolic inference, such as RETOMATON (Alon et al., 2022).

To reduce the computation cost of finetuning large-scale LMs, and the storage cost of the finetuned models, PEFT methods only update partial parameters. Compared with full parameter finetuning, PEFT methods can better avoid catastrophic forgetting (Goodfellow et al., 2013), and learn better from a small amount of data. Adapter methods directly introduce new modules to the LMs, such as bottleneck adapters (Houlsby et al., 2019) and compacter (Davison, 2021). PROMPT TUNING prepends tunable tokens to the input data. Similarly, PREFIX-TUNING modifies the multi-head attention with new parameters, that is, prepending trainable vectors to the key and value matrices. LORA (Hu et al., 2021) uses low-rank matrix to learn the additive updates on the self-attention weight matrix, while its variants are improving the parameter efficiency, such as ADALORA (Zhang et al., 2023), DORA (yang Liu et al., 2024). Similarly, OPT (Qiu et al., 2023) uses orthogonal transformations to learn the multiplicative weight updates, and its variant BOFT (Liu et al., 2023) introduces factorization techniques for better parameter efficiency. While IA$^3$ (Liu et al., 2022) rescales the activations by tuning additional coefficients. In addition, some work is the combinations of other PEFT methods (Mao et al., 2021). For example, mix-and-match adapters are a combination of prefix-tuning and bottleneck adapters (He et al., 2021).

## 7 CONCLUSION

In this paper, we proposed a novel approach for in-context learning, "vocabulary-defined semantics". It incorporates KNN decisions into LM inference to improve the performance, and also, makes full use of all usable samples to avoid the process of demonstration selection. We define semantic bases, a collection of semantically equivalent representations to label space of language model, to define the semantic property of LM latent space. Leveraging the local isotropy of latent space, we propose similarity-based logits to quantify the semantic gap of the language model with downstream data. Further, we introduce a neural clustering module to optimize the logits, which in turn mitigates the semantic gap. Based on the results of extensive experiments, our semantic-based approach shows effectiveness and efficiency LM adaptation. It significantly outperforms the state-of-the-art, while showing advantages in both the computation cost and storage cost. Moreover, through our ablation study, similarity-based logits is as good as the multiplication-based logits computation, but performs better in numerical robustness. In the future, we will explore more topics where our semantic-based approach can help, contributing to not only LM performance, but also LM interpretability.

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

# A    IMPLEMENTATION DETAILS

## A.1    PROMPT TEMPLATES

The prompt templates follow the practice of prior work (Lu et al., 2021), as shown in Table 7.

## A.2    HYPERPARAMETERS

In the demonstration retrieval, we randomly sample data from the training set. Following the practice of KP, we compute the maximum allowed number of demonstrations by setting the threshold of the truncation probability to be $5\%$. We stabilize the experiments using the default random seed $42$.

In the clustering process of our approach VDS, since the neural clustering module is outside language models and directly working on the detached latent representations, we take large hyperparameters to speedup semantic clustering: the epoch number is $100$ and the batch size is $256$.

In the learning process of PEFT methods, we follow the recommended practices of LoRA and IA$^3$ based on their papers. That is, making the query-key-value matrices trainable in LoRA, and also making the second fully-connected layer in the feed-forward module trainable in IA$^3$. We take the commonly used hyperparameters, keeping the epoch number $1$ and the batch size $1$.

# B    MORE DISCUSSION

## B.1    GENERAL SEMANTIC BASIS

Considering the semantic basis is based on the LM vocabulary, we can duplicate the practice on the LM last-layer (after the last-layer, before LM-head) to the LM embedding-layer (after the embedding layer, before the first-layer) as well. For the sake of the opposite computation direction

| Dataset | Prompt | Label Space |
|---|---|---|
| AGNews | Input: Nets get Carter from Raptors. "INDIANAPOLIS – All-Star Vince Carter was traded by the Toronto Raptors to the New Jersey Nets for Alonzo Mourning, Eric Williams, Aaron Williams, and a pair of first-round draft picks yesterday." 
 Type: business | world, sports, business, technology |
| CARER | message: i know a lot but i feel so stupid because i can not portray it 
 emotion: sadness | sadness, joy, love, anger, fear, surprise |
| MR | review: "provides a porthole into that noble , trembling incoherence that defines us all ." 
 sentiment: positive | negative, positive |
| MRPC | Premise: The 30-year bond US30YT = RR rose 22 / 32 for a yield of 4.31 percent , versus 4.35 percent at Wednesday 's close . 
 Hypothesis: The 30-year bond US30YT = RR grew 1-3 / 32 for a yield of 4.30 percent , down from 4.35 percent late Wednesday . 
 Prediction: not equivalent | equivalent, not equivalent |
| SST5 | review: a deliciously nonsensical comedy about a city coming apart at its seams . 
 sentiment: great | terrible, bad, okay, good, great |
| SUBJ | input: "looking for a short cut to fame , glass concocted sources , quotes and even entire stories , but his deception did not go unnoticed forever , and eventually , his world came crumbling down . . ." 
 type: objective | subjective, objective |
| TREC | Question: What currency is used in Australia ? 
 Type: entity | description, entity, expression, human, location, number |
| WebSS | input: americangymnasticsclub american gymnastic club recreational gymnastics boys girls schedule fees programs calendar birthday parties camps staff 
 type: sports | business, computers, culture-arts-entertainment, education-science, engineering, health, politics-society, sports |

Table 7: Prompt Template and Label Space in the Experiments.

of representation, the semantic basis can be obtained by a matrix multiplication between onehot embedding and the embedding matrix. That is, we multiply the onehot embedding $\vec{e}$ by the embedding matrix $\mathbb{W}$ to obtain the corresponding representation $\vec{r}$. Expressed in formula, that is, $\vec{r} = \vec{e} \cdot \mathbb{W}$.

### B.2 DISENTANGLEMENT IN LATENT SPACE

The insights motivating vocabulary-defined semantics is logits disentanglement. In our approach, we compute logits before the LM-head, instead of after. Therefore, we can obtain the disentangled logits in latent space. In the latent space, the disentangled logits are useful to evaluate and optimize latent representations. Our approach is leveraging the vocabulary to define the semantics of last-layer latent space, therefore, we mainly guarantee the usefulness in the LM last-layer.

## C MORE RESULTS AND ANALYSIS

### C.1 QUALITY OF LATENT SPACE CLUSTERING

We can measure the quality of representation clustering via clustering metrics, such as Adjusted Rand Index (ARI). The computation mechanism revolves around pairwise comparisons on clusterings. ARI measures how similar two clusterings are in each comparison, while accounting for the possibility of random agreement. The value range is $[-1, 1]$, the larger, the better. A value $0$ or values close to $0$ indicate a normal disordered situation.

The results are shown in Table 8, where we use labels in the form of `metric[identifier]` to mark variants, and identifier is either *w/o* or *w/*, representing whether semantic clustering is applied. We can see that semantic clustering can bring obvious improvements in the clustering quality.

### C.2 BENEFITS OF SEMANTIC CLUSTERING TO KNN DECISION

Since the clustering quality is promoted by semantic clustering, we can anticipate that, the performance of the KNN decision will also be benefited. That is, our approach incorporates an improved KNN decision process into LM inference. For ease of description, we use "sibling" to describe data

| Dataset | Metric | GPT2 | | | | Qwen2 | | | Gemma2 | | Llama3 |
|---|---|---|---|---|---|---|---|---|---|---|---|
| | | Small | Medium | Large | XL | 0.5B | 1.5B | 7B | 2B | 9B | 8B |
| AGNews | ARI[w/o] | 0.306 | 0.285 | 0.351 | 0.265 | 0.145 | 0.425 | 0.528 | 0.000 | 0.000 | 0.402 |
| | ARI[w/] | 0.798 | 0.811 | 0.822 | 0.822 | 0.764 | 0.844 | 0.847 | 0.772 | 0.773 | 0.851 |
| CARER | ARI[w/o] | 0.015 | 0.015 | 0.033 | 0.096 | 0.153 | 0.135 | 0.236 | 0.037 | 0.018 | 0.172 |
| | ARI[w/] | 0.426 | 0.407 | 0.377 | 0.430 | 0.595 | 0.485 | 0.539 | 0.458 | 0.412 | 0.429 |
| MR | ARI[w/o] | 0.049 | 0.041 | 0.382 | 0.316 | 0.245 | 0.179 | 0.565 | 0.000 | 0.029 | 0.416 |
| | ARI[w/] | 0.384 | 0.502 | 0.514 | 0.536 | 0.494 | 0.649 | 0.734 | 0.224 | 0.203 | 0.659 |
| MRPC | ARI[w/o] | 0.000 | 0.000 | 0.000 | 0.000 | 0.000 | -0.002 | 0.000 | 0.000 | 0.000 | -0.002 |
| | ARI[w/] | 0.114 | 0.110 | 0.119 | 0.096 | 0.098 | 0.269 | 0.369 | 0.106 | 0.088 | 0.139 |
| SST5 | ARI[w/o] | 0.022 | 0.018 | 0.004 | 0.009 | 0.052 | 0.030 | 0.244 | 0.004 | 0.000 | 0.018 |
| | ARI[w/] | 0.106 | 0.162 | 0.154 | 0.169 | 0.162 | 0.192 | 0.247 | 0.054 | 0.072 | 0.196 |
| SUBJ | ARI[w/o] | 0.001 | 0.121 | 0.013 | 0.000 | 0.028 | 0.025 | 0.087 | 0.009 | 0.030 | 0.194 |
| | ARI[w/] | 0.684 | 0.714 | 0.741 | 0.792 | 0.727 | 0.817 | 0.841 | 0.621 | 0.656 | 0.848 |
| TREC | ARI[w/o] | 0.100 | 0.074 | 0.151 | 0.070 | 0.292 | 0.083 | 0.138 | 0.003 | 0.000 | 0.183 |
| | ARI[w/] | 0.767 | 0.837 | 0.836 | 0.879 | 0.826 | 0.882 | 0.907 | 0.752 | 0.761 | 0.862 |
| WebSS | ARI[w/o] | 0.171 | 0.202 | 0.207 | 0.210 | 0.050 | 0.104 | 0.214 | 0.050 | 0.019 | 0.191 |
| | ARI[w/] | 0.578 | 0.606 | 0.590 | 0.620 | 0.579 | 0.638 | 0.667 | 0.475 | 0.457 | 0.666 |

Table 8: LM Performance w/ and w/o Semantic Clustering.

| Dataset | Parameter | GPT2 | | | | Qwen2 | | | Gemma2 | | Llama3 |
|---|---|---|---|---|---|---|---|---|---|---|---|
| | | Small | Medium | Large | XL | 0.5B | 1.5B | 7B | 2B | 9B | 8B |
| AGNews (Acc.) | k=1[w/o] | 0.704 | 0.721 | 0.759 | 0.760 | 0.578 | 0.721 | 0.739 | 0.576 | 0.514 | 0.795 |
| | k=1[w/] | 0.915 | 0.918 | 0.925 | 0.928 | 0.898 | 0.935 | 0.938 | 0.909 | 0.898 | 0.940 |
| | k=16[w/o] | 0.819 | 0.816 | 0.846 | 0.845 | 0.703 | 0.833 | 0.851 | 0.748 | 0.708 | 0.881 |
| | k=16[w/] | 0.919 | 0.924 | 0.929 | 0.929 | 0.904 | 0.937 | 0.939 | 0.908 | 0.908 | 0.941 |
| | k=256[w/o] | 0.852 | 0.855 | 0.876 | 0.875 | 0.760 | 0.861 | 0.886 | 0.825 | 0.817 | 0.899 |
| | k=256[w/] | 0.919 | 0.924 | 0.929 | 0.929 | 0.903 | 0.938 | 0.939 | 0.909 | 0.908 | 0.941 |
| WebSS (F1) | k=1[w/o] | 0.239 | 0.273 | 0.397 | 0.372 | 0.326 | 0.329 | 0.282 | 0.220 | 0.202 | 0.359 |
| | k=1[w/] | 0.682 | 0.703 | 0.694 | 0.702 | 0.688 | 0.718 | 0.723 | 0.632 | 0.636 | 0.726 |
| | k=16[w/o] | 0.345 | 0.413 | 0.592 | 0.580 | 0.493 | 0.506 | 0.465 | 0.290 | 0.249 | 0.537 |
| | k=16[w/] | 0.691 | 0.711 | 0.701 | 0.709 | 0.695 | 0.736 | 0.728 | 0.638 | 0.653 | 0.730 |
| | k=256[w/o] | 0.560 | 0.601 | 0.738 | 0.730 | 0.634 | 0.686 | 0.638 | 0.468 | 0.362 | 0.709 |
| | k=256[w/] | 0.796 | 0.803 | 0.794 | 0.804 | 0.787 | 0.818 | 0.837 | 0.730 | 0.727 | 0.831 |

Table 9: LM Performance with $k$ Nearest-Neighbor Methods.

representations that correspond to the same label in the vocabulary, and use "neighbor" to describe data representations that are close to each other in latent space. In semantic clustering, for each data, its siblings shall gradually become its neighbors.

We experimented with semantic clustering to validate its effects on nearest-neighbor methods, that is, predicting the class of data representations with the reference to the nearest neighboring data in the latent space. As shown in Table 9, there are obvious improvements in the performance when we apply semantic clustering. When the neighboring number $k$ increased from $k = 1$ to $k = 256$, the performance of [w/o] is gradually increased and becomes closer to that of [w/]. The accuracy changes shown on [w/o] mean, that semantic clustering transforms the representations to make more neighboring data be the sibling data, which fulfills the expectation.

# D  REPRESENTATIONS IN LATENT SPACE

To illustrate the effects of semantic clustering, we visualize the representations in the latent space using tSNE (Arvanitidis et al., 2017), mainly on the AGNews dataset. We conclude findings from the illustrations: (1) concerning the distribution of latent representations, the level of representations confounding with each other is well reduced by clustering; (2) the representations distribute differently in the latent space of different models. However, the effects of semantic clustering in these latent spaces are almostly similar, as shown in Figure 2, Figure 3, and Figure 5; (3) the distribution of latent representations before clustering will affect the distribution after clustering, especially when the confounding level is very high, such as the case of GEMMA2-9B, shown in Figure 4.

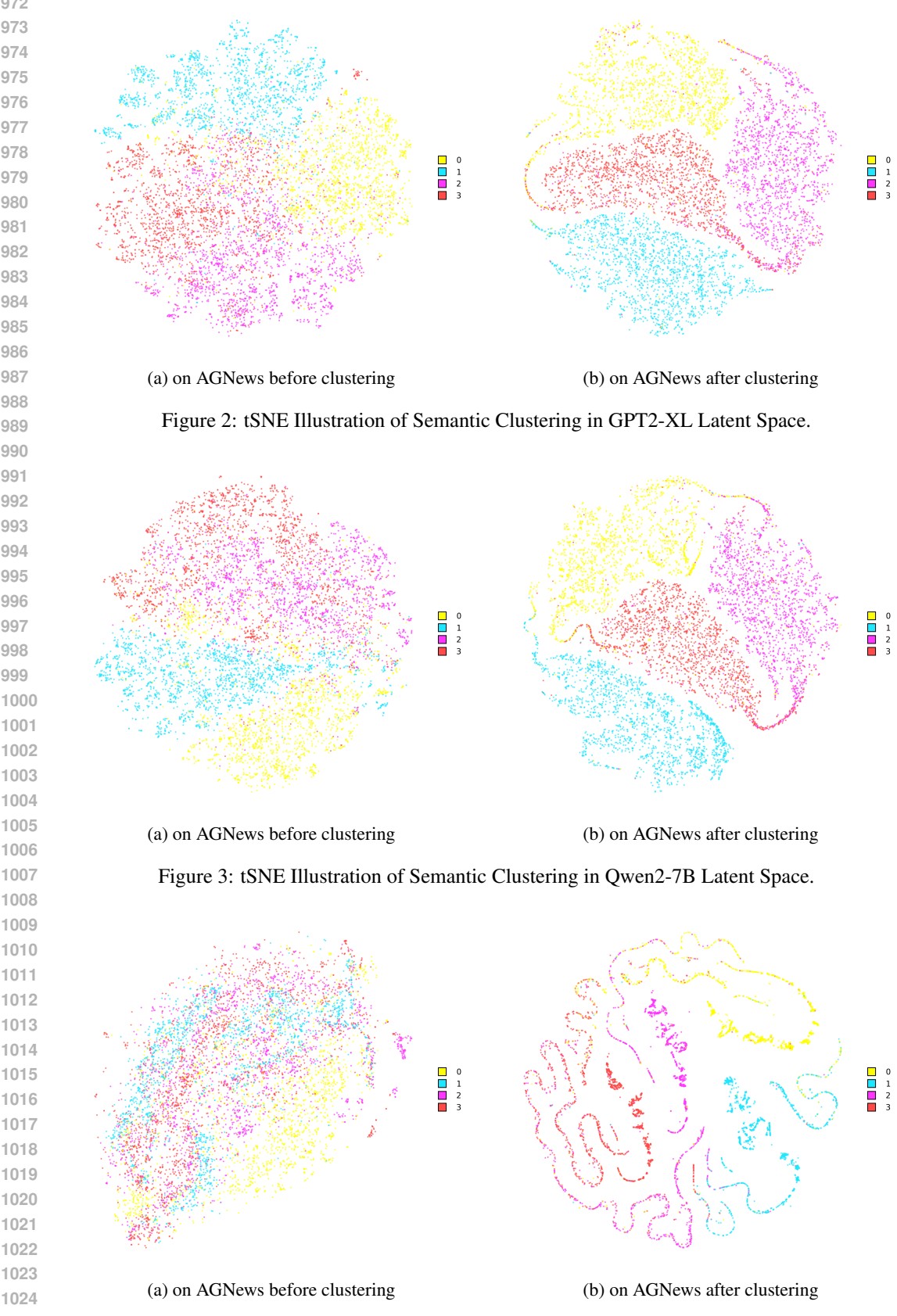

(a) on AGNews before clustering      (b) on AGNews after clustering

Figure 2: tSNE Illustration of Semantic Clustering in GPT2-XL Latent Space.

(a) on AGNews before clustering      (b) on AGNews after clustering

Figure 3: tSNE Illustration of Semantic Clustering in Qwen2-7B Latent Space.

(a) on AGNews before clustering      (b) on AGNews after clustering

Figure 4: tSNE Illustration of Semantic Clustering in Gemma2-9B Latent Space.

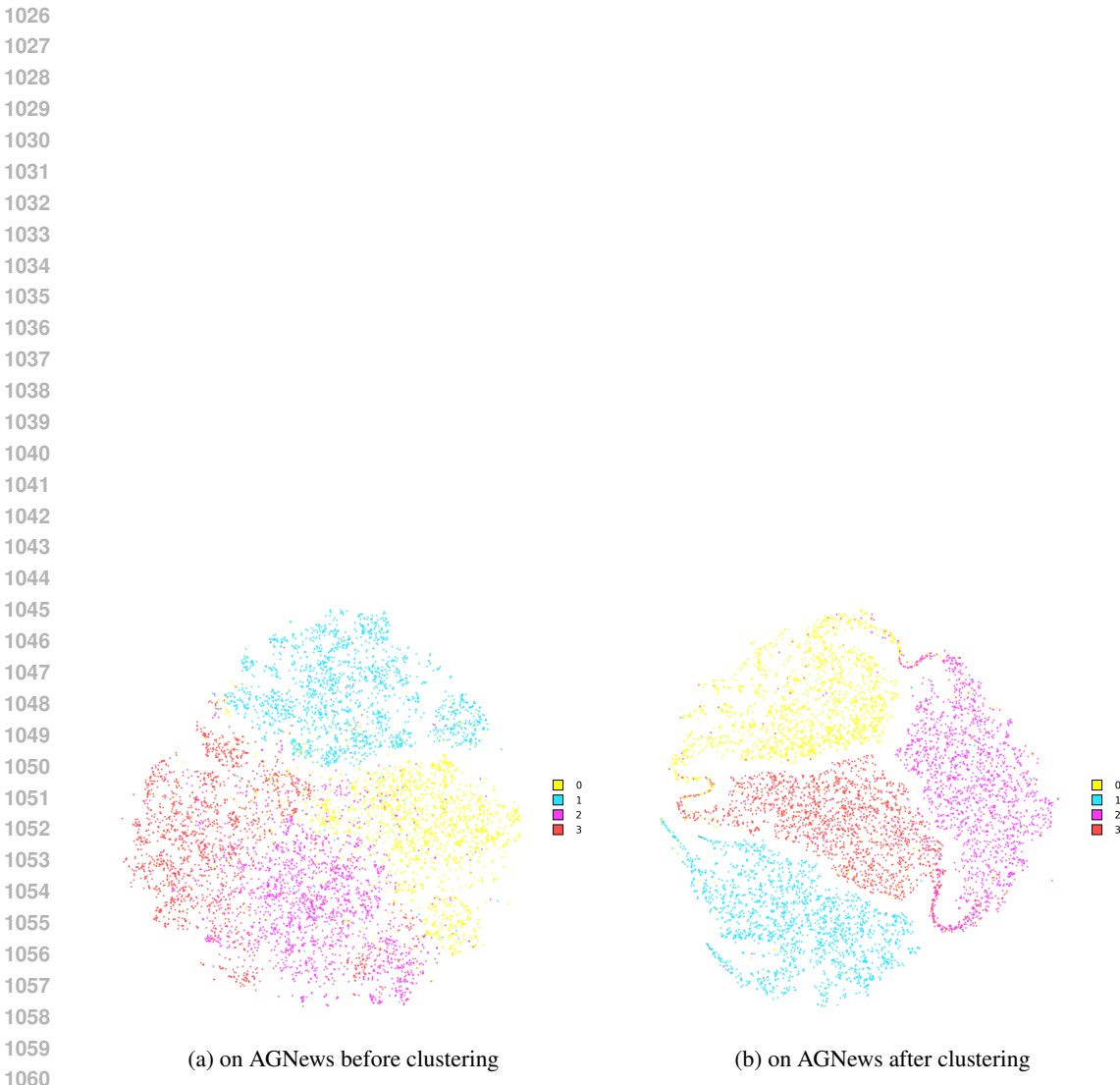

(a) on AGNews before clustering                    (b) on AGNews after clustering

Figure 5: tSNE Illustration of Semantic Clustering in Llama3-8B Latent Space.

