# OpenReview forum: "Vocabulary-Defined Semantics: Latent Space Clustering for Improving In-Context Learning"
_ICLR.cc/2025/Conference — Submitted to ICLR 2025_

### Official Review · Reviewer_dDgL · 2024-10-19

**Soundness:** 3
**Presentation:** 1
**Contribution:** 2
**Rating:** 3
**Confidence:** 4

**Summary:**

Summary:

This work proposes a new demonstrations retrieval method in in-context learning by utilizing the latent space of language models.

Contributions:

The proposed method can outperform traditional random in-context learning and the other work named KP.

**Strengths:**

1. The proposed method can outperform traditional random in-context learning and the other work named KP.
2. This work explores the possibilities of utilizing the latent space of language models to better understand the downstream tasks.

**Weaknesses:**

1. The experiment section is quite insufficient. To begin with, it is widely recognized that numerous methods exist to enhance the quality of demonstration retrieval in in-context learning. However, these methods have not been evaluated or listed for comparative purposes. Additionally, the selection of a weak baseline—traditional in-context learning with randomly retrieved demonstrations—raises concerns about the robustness of the findings. The absence of comparative experiments leads to apprehension regarding potential overclaims made by the authors.

2. The related work section is inadequately developed. As previously noted, there exists a substantial body of literature addressing demonstration retrieval in in-context learning that has not been referenced or discussed within this section.

3. Furthermore, it is important to highlight that all datasets employed in this study pertain exclusively to classification tasks, with no inclusion of text-generation tasks. This significant limitation should be explicitly acknowledged in the paper.

4. Last but not least, the writing of this paper can be quite aweful. The authors fail to explain the new terminology which confuses me a lot. Additionally, the references section does not meet academic standards, as it appears to rely solely on citations from Semantic Scholar.

**Questions:**

1. I doubt whether your method can help Gemma2 so much. The poor results of Gemma2 of KP and ICL with randomly selected demonstrations have implied that it even fails to understand the task. Under this condition, I have no idea why the latent space of Gemma2 can help.

---

> ### Author Response · Authors · 2024-11-22
>
> We thank the reviewer for your comments. We want to clarify a few points regarding our work as follows:
>
> **For Weakness#1 and Weakness#3:**
>
> The focus of our work is more focused on *beyond-context learning*. It addresses the scalability problem of in-context learning due to the context length restriction, where the performance cannot scale with the amount of available training examples. This is the reason why we mainly compare our approach with Knn-Prompting, and also, conduct experiments on text classification tasks following the experimental setups of Knn-Prompting. To the best of our knowledge, Knn-Prompting is the state-of-the-art in beyond-context learning. We will better clarify the focus of our paper in the revised version.
>
> **For Weakness#2:**
>
> We see that the related work section does not involve all of the recent progress of ICL. Since our focus is more on *beyond-context learning*, in related work, we discuss more on the related techniques in overcoming the context length restrictions of in-context learning, as well as the model finetuning techniques. We will better clarify the focus of our paper in the revised version, and also, include more recent work on ICL.
>
> **For Weakness#4:**
>
> There is only one new term in our paper, that is “semantic basis”. As defined in section 3, it means the semantically equivalent representation in LM latent space, corresponding to any given label on LM vocabulary. We guess your feeling of confusion is due to its concept being hard to imagine. Our illustrations in Figure 1 are beneficial to clarify the concept of “semantic basis”, and also, it shall be helpful to think with the math formula (line 153).
>
> **For Question#1:**
>
> Our approach can improve the quality of latent space, so when the initial latent space is poor (such as Gemma2), our approach shows significant improvements. The reason why KP and ICL cannot perform well is because the predicted probabilities of the ground truth are not high enough, that is, the latent representations are not close enough to the corresponding ground truth. As a countermeasure, our clustering approach can move latent representations closer to the corresponding ground truth, so the logits become better (the predicted probabilities of the ground truth become higher).

---

> ### Author Response · Authors · 2024-11-28
>
> Dear reviewer, we have made a revision on our paper based on your comments, please check. Thanks!

---

### Official Review · Reviewer_EKjp · 2024-10-30

**Soundness:** 2
**Presentation:** 1
**Contribution:** 1
**Rating:** 1
**Confidence:** 1

**Summary:**

This paper proposes a 'latent space clustering' method to improve ICL performance, which is named as VDS (Vocabulary-Defined Semantics). With the incorporation of learnable parameters, this method is reported to achieve significant improvements over vanilla ICL and KNN prompting. A comparison with other PEFT techniques is also presented. While the performance does not surpass IA in all respects, VDS demonstrates notable advantages in terms of reduced training and inference costs. In the conclusion section, the authors suggest that their method "outperforms the state-of-the-art," with additional benefits in computational efficiency and storage requirements.

**Strengths:**

1. This paper studies In-context Learning, which is an important problem.
2. The authors claim that their method "outperforms the state-of-the-art," while showing advantages in both the computation cost and storage cost.
3. The authors have succeeded in making their work intellectually stimulating, encouraging readers to exercise their analytical skills to decode the presented concepts.

**Weaknesses:**

1. VDS requires learnable parameters like IA$^3$ but shows inferior performance in all datasets.
2. This method, thanks to the author's description, is sophisticated to understand and implement.

**Questions:**

Necessary justification should be provided to explain why this "VDS" method should presented in an ineffable manner.

---

### Official Review · Reviewer_vhaJ · 2024-11-01

**Soundness:** 3
**Presentation:** 2
**Contribution:** 3
**Rating:** 5
**Confidence:** 3

**Summary:**

The paper proposes a novel in-context learning method by replacing the indexing operation of KNN prompting with a clustering operation to align the semantic properties between the language model and the downstream data/tasks.  The model constructs semantic bases to represent the label space of model output and introduces similarity-based logits to quantify the semantic gap of the language model with downstream data. Furthermore, the approach innovates by introducing a centroid-known clustering module to optimize the logits, mitigating the semantic gap.

**Strengths:**

Adopting similarity-based semantic logits with the help of semantic bases to quantify the semantic gap is interesting.

**Weaknesses:**

1. The authors only compare the proposed method with two baselines in the experiment. However, many in-context learning methods have been proposed in recent two years. The baselines are insufficient.

2. The necessity of similarity-based logits is not particularly convincing. It can be observed in Tab.6 that the improvements brought by sm are limitied.

3. The authors claim that the proposed model is dependent on the quality of latent representations, but this characteristic has not been validated through relevant experiments.
4. The ablations performs in the paper are unclear. Although some component-level ablations on the model have been implemented, it's unclear what the take-away message from ablations is.

**Questions:**

1.The results in Table 3 show that the improvement of the proposed method compared to ICL and KP is significant on the CARER, TREC, and WebSS datasets. Can the authors further analyze the reasons for this?

2. I wonder whether the similarity-based logit computation or the clustering module contributes more to the final performance.

---

> ### Author Response · Authors · 2024-11-22
>
> We thank the reviewer for your comments. We want to clarify a few points regarding our work as follows:
>
> **For Weakness#1:**
>
> The focus of our work is more focused on *beyond-context learning*. It addresses the scalability problem of in-context learning due to the context length restriction, where the performance cannot scale with the amount of available training examples. This is the reason why we mainly compare our approach with Knn-Prompting. To the best of our knowledge, Knn-Prompting is the state-of-the-art in beyond-context learning. We will better clarify the focus of our paper in the revised version.
>
> **For Weakness#2:**
>
> The similarity-based logits computation is essential in supporting the clustering operation, since the logits play the role of the clustering objective. Meanwhile, in our analysis in section 5, the similarity-based logits can significantly save the computation cost (by several orders of magnitude), while maintaining the same performance.
>
> **For Weakness#3:**
>
> Due to the limitation of pages, the study on the quality of latent representations is in section C of the appendix. There, we reported the Adjusted Rand Index (ARI), a clustering metric, to measure the quality of latent representations concerning KNN decision (see section C.1). The clustering metric indicates the benefits of introducing KNN decision to the LM inference. The quality of the original latent representations was bad so the benefit of KNN decision was marginal (see section C.2) before clustering. After clustering, the quality of latent representations is improved, and so is the benefit of KNN decision.
>
> **For Weakness#4:**
>
> The take-away messages from the ablation study are: (1) in model inference, similarity-based logits computation is equivalent to the general logits computation; (2) in clustering, similarity-based logits computation can be more efficient while maintaining similar effectiveness; (3) in disentanglement, the similarity-based logits computation shows better robustness in maintaining the performance.
>
> **For Question#1:**
>
> Regarding these three datasets, the main factor deciding the improvements of our approach is “the number of classes”. The stats of the mentioned datasets (CARER: 6, TREC: 6, and WebSS: 8) are larger than others. Meanwhile, a larger number of classes indicates the difficulty of distinguishing the latent representations without additional interventions (such as clustering), and also, suggests the potential benefits of using semantic-clustering for improving the quality of latent representation. Meanwhile, the difficulty of data/tasks and the capability of models may affect the improvements of our approach as well.
>
> **For Question#2:**
>
> Both are important to the final performance. The similarity-based logits computation contributes more to the efficiency (greatly reducing the computation cost), and the clustering module contributes more to the effectiveness (realizing 3%-49% improvements).

---

> ### Author Response · Authors · 2024-11-28
>
> Dear reviewer, we have made a revision on our paper based on your comments, please check. Thanks!

---

### Official Review · Reviewer_GG8H · 2024-11-04

**Soundness:** 2
**Presentation:** 3
**Contribution:** 2
**Rating:** 5
**Confidence:** 3

**Summary:**

The paper addresses instability issue in in-context learning (ICL) method for language models due to inconsistent quality, format, or order of prompts. It proposes a novel method called "vocabulary-defined semantics" to enhance ICL by clustering semantically equivalent latent representations based on model output labels. This approach aims to optimize demonstration selection and align semantic properties between language models and downstream tasks. Experiments show that the proposed method outperforms state-of-the-art approaches, achieving 3% to 49% improvement in performance while reducing computation time by half.

**Strengths:**

1. Paper Presentation: The paper is clearly structured and systematically presents the proposed method, making it easy for readers to follow the motivation, methodology, and contributions. The discussion of related work and limitations of existing methods helps set up the context effectively, while the proposed solution is logically introduced.

2. Promising Experiment Results in Classification Task: The experimental setup is comprehensive, utilizing eight diverse text understanding datasets for classification tasks, which demonstrates the robustness and generalizability of the approach. The performance improvement ranges from 3% to 49%, highlights the effectiveness of the method. The comparisons include both effectiveness and efficiency, showing significant performance gains while reducing computation time.

**Weaknesses:**

1. Limited Evaluation Scope: The method is only evaluated in text classification tasks, despite in-context learning (ICL) being widely used in text generation tasks, such as question answering (QA). Evaluating the approach on these broader tasks could provide a more comprehensive understanding of its effectiveness.

2. Need for More Baseline Comparisons: The paper could benefit from including additional baseline methods, especially considering that ICL methods have been widely studied with different approaches, such as self-reflective retrieval or feedback-augmented ICL.

3. Ablation Study Lacks Analysis of Semantic Basis: The ablation study does not analyze the choice of the semantic basis. It would be beneficial to explore whether using a larger or smaller basis impacts the performance, as this could provide insights into the scalability and flexibility of the proposed method.

**Questions:**

1. Are there any limitations in applying this method to text generation tasks?
2. How can we determine the appropriate semantic basis, or are there any insights into how the choice of semantic basis affects performance?

---

> ### Author Response · Authors · 2024-11-22
>
> We thank the reviewer for your comments. We want to clarify a few points regarding our work as follows:
>
> **For Weakness#1 and Weakness#2:**
>
> The focus of our work is more focused on *beyond-context learning*. It addresses the scalability problem of in-context learning due to the context length restriction, where the performance cannot scale with the amount of available training examples. This is the reason why we mainly compare our approach with Knn-Prompting, and also, conduct experiments on text classification tasks following the experimental setups of Knn-Prompting. To the best of our knowledge, Knn-Prompting is the state-of-the-art in beyond-context learning. We will better clarify the focus of our paper in the revised version.
>
> **For Weakness#3:**
>
> The semantic basis means the semantic-equivalent latent representation concerning each label/token in LM vocabulary. In other words, for each ground truth, there is only one determined semantic basis. Therefore, the ablation study can only be conducted on the clustering module, not the semantic basis.
>
> **For Question#1:**
>
> No limitations. The method uses a semantic-based clustering module to transform latent representations.
>
> **For Question#2:**
>
> The semantic basis is the pseudo-inverse results between the LM-head matrix and one-hot probabilities. For each label/token in LM vocabulary, there is a corresponding semantic basis, which is unique when ignoring the error of calculation accuracy. Therefore, the choice of semantic basis depends on what is the needed label/token.

---

> ### Author Response · Authors · 2024-11-28
>
> Dear reviewer, we have made a revision on our paper based on your comments, please check. Thanks!

---

### Meta-Review · Area_Chair_9jZi · 2024-12-17

**Metareview:**

The paper tackles instability in in-context learning (ICL) caused by inconsistent prompt quality, format, or order. It introduces a novel method, "vocabulary-defined semantics" (VDS), which clusters semantically equivalent latent representations to optimize demonstration selection. This approach aligns the semantic properties of language models with downstream tasks, enhancing both accuracy and efficiency. Experiments on text classification tasks show that the proposed method achieves performance improvements of 3% to 49% over baseline methods while significantly reducing computation time.

Strengths

+ Comprehensive Experiments: The approach is tested on eight diverse text classification datasets, demonstrating robustness and generalizability.
+ Efficient and Cost-effective: The method provides computational and storage advantages over other state-of-the-art techniques.

Weaknesses

+ Limited Evaluation Scope: The method is only evaluated on classification tasks, neglecting broader tasks like text generation or question answering.
+ Insufficient Baseline Comparisons: The paper compares the proposed method to only a few baselines, overlooking many recent advancements in ICL methods.
+ Ablation Study Limitations: The analysis lacks clarity and depth, particularly regarding the necessity and choice of semantic bases and similarity-based logits.
+ Related Work and Writing Issues: The related work section is underdeveloped, and the paper contains confusing terminology and poorly structured writing.
+ Performance Gaps: While claiming state-of-the-art improvements, the proposed method shows inferior results compared to certain advanced baselines like IA in some datasets.

Some concerns have been addressed by the authors during the rebuttal period.

**Additional Comments On Reviewer Discussion:**

This paper received all-negative ratings. All reviewers agree that there are major issues in this paper that need to be addressed, especially in terms of baselines and evaluation in general. The authors provide some response. I have a hard time encouraging reviewers to engage in the discussion, but it appears that the rebuttal did not convince the reviewers.

---

### Decision · Program_Chairs · 2025-01-22

Reject